# Effect of Heat Stress on Subsequent Estrous Cycles Induced by PGF2α in Cross-Bred Holstein Dairy Cows

**DOI:** 10.3390/ani14132009

**Published:** 2024-07-08

**Authors:** Passawat Thammahakin, Adisorn Yawongsa, Theera Rukkwamsuk

**Affiliations:** 1Laboratory of Public Health, Faculty of Veterinary Medicine, Hokkaido University, Sapporo 060-0818, Japan; passawat.th@vetmed.hokudai.ac.jp; 2Department of Large Animal and Wildlife Clinical Sciences, Faculty of Veterinary Medicine, Kasetsart University, Nakhon Pathom 73140, Thailand; fvetady@ku.ac.th

**Keywords:** heat stress, dairy cow, rectal temperature, respiratory rate, reproduction

## Abstract

**Simple Summary:**

Heat stress (HS) influenced reproductive performance and physiological responses in cross-bred Holstein dairy cows under tropical conditions. First-service conception rates of cows under moderate heat stress tended to be lower than those under low heat stress. The average interval between calving and first service in cows under moderate heat stress was longer than that under mild heat stress. Cows in moderate heat stress responded less to PGF2α than cows in mild heat stress. This study indicated that heat stress influenced the reproductive performance of cows raised in Thailand throughout the year, especially when heat stress was relatively intense.

**Abstract:**

This study aimed to determine the effect of heat stress (HS) on reproductive parameters (calving to first service (CTFS) and the first-service conception rate (FSCR)) and general physiological responses (rectal temperature (RT) and respiratory rate (RR)) in tropical cross-bred Holstein dairy cows raised in Ratchaburi province, Thailand. HS was determined using the temperature–humidity index (THI), calculated from temperature and humidity inside the barns, and was classified as moderate HS (THI: 80.67 ± 0.79) and mild HS (THI: 77.81 ± 1.09) in this study. Cows with detected corpus luteum were defined as cyclic cows and were injected with PGF2α at the beginning of the experimental period. Reproductive and physiological parameters were recorded. Cows showed significantly lower RT and RR in the mild HS group (38.47 ± 0.21 °C and 41.04 ± 4.55 bpm, respectively) than in the moderate HS group (38.87 ± 0.15 °C and 51.17 ± 10.52 bpm). The percentage of cows that ovulated after being induced by PGF2α and showed estrus signs was higher in the mild than the moderate HS groups (54.55% vs. 18.18%). Furthermore, the FSCR of cows under mild HS tended to be higher than that in the moderate HS group (42.11% and 15%, respectively) (*p* = 0.06), while the average CTFS interval was significantly shorter under mild HS than moderate HS (69.47 ± 18.18 and 84.60 ± 27.68 days, respectively). These results indicate that moderate HS impairs reproductive performance in crossbred Holstein cows, compared to mild HS conditions.

## 1. Introduction

Global warming has been increasing worldwide, including in Thailand, a tropical country with high ambient temperatures and relative humidity for most of the year. Between 2006 and 2011, the highest average temperature in the western part of Thailand was 32.6 °C, which is higher than a cow’s thermoneutral zone and could cause heat stress (HS) due to undissipated body heat [1].

HS can be detected by the temperature–humidity index (THI), a single value representing the combined effects of environmental temperature and relative humidity. The comfort zone of many animals is a THI lower than 71. Values between 72 and 79 imply the potential for mild HS, between 80 and 89 are considered moderate HS, and above 90 are considered severe HS for animals [2]. On average, in Thailand, a high THI is present for eight months of the year (March to October), with durations of 9 h/day (from 10 a.m. to 7 p.m.) [3]. Ratchaburi is a province located in western Thailand with high dairy farm density [4]; cows raised in this area are likely to face mild to moderate HS.

HS has a severe impact on health performance and the reproductive system. Cows can adapt to HS by quickly dissipating heat by increasing their respiratory rate (RR) [5]. HS also affects feed intake capacity [6,7,8] and reduces rumen activity [9]. Consequently, it reduces body condition scores and could aggravate negative energy balance during postpartum [10]. HS has various adverse effects on milk production, including yield and composition [11,12], and it can affect the body’s immune system [13,14]. Furthermore, HS is a significant factor in poor reproductive performance [15,16]. Several studies have reported that HS could alter various aspects of the hypothalamic–pituitary–ovarian axis, especially in follicular growth [6,17,18]. HS induced apoptosis of granulosa cells and resulted in poor follicle quality [19,20]. Granulosa cells are the primary cells that produce estrogen and inhibin concentrations. Thus, HS reduced steroidogenesis [6,17,18,19] and decreased estrous behavior and silent heat [21]. Less estrogen was also caused by positive and negative feedback related to the reduced pre-ovulatory follicle dominance process, delayed ovulation, and formation of poor-quality luteal cells, which altered progesterone hormone [22]. Hence, these adverse effects (especially during the hot season) led to low reproductive performance in Thailand, including delayed postpartum ovulation, prolonged days open, and low first-service conception rates (FSCRs) [23,24,25].

Evaluating general physiological responses associated with the effect of HS can improve our understanding and confirm the severity of HS in cows raised in Thailand. In practice, Thai bovine practitioners prefer to use hormonal treatment, most likely PGF2α, to deal with low reproductive performance in dairy cows. However, the success rate of this hormonal treatment in those cows is uncertain, which was postulated to be linked to HS during the treatment period. Furthermore, the effect of HS on reproductive parameters and general physiological responses is very limited in field studies in Thailand. Therefore, the results obtained from this study could help develop and improve the farm management of heat-stressed cows. In addition, this information could be used as a reference for the reproductive index of cows under HS conditions, and practitioners could apply the results to improve hormone administration protocols for reducing reproduction problems due to HS in dairy cows.

Therefore, this research focused on the effect of HS on reproductive parameters and general physiological responses during subsequent estrous cycles induced by PGF2α in tropical Holstein Friesian dairy cows in Ratchaburi province in the western part of Thailand.

## 2. Materials and Methods

This study was approved by the Committee for Animal Ethics in Scientific Procedures of Kasetsart University, Bangkok, Thailand (Approval Number: ACKU62-VET-047).

### 2.1. Daily Temperature and Humidity Data

Air temperature and relative humidity in the barn were recorded every three hours using three data loggers (Ibutton^®^, model: DS1923, Maxim Integrated, San Jose, CA, USA) throughout the experiments. These data were transferred monthly through the 1-Wire^®^ program and exported to a .CSV file (Microsoft® Excel version 365). The temperature humidity index (THI) was calculated according to the following formula: THI = (0.8 T) + ((RH/100) × (T − 14.4)) + 46.4, where T is the ambient temperature of the air (°C) and RH is the relative humidity (%) [3]. The average monthly THI ranged between 81.89 and 76.64 between April 2018 and January 2019.

### 2.2. Animals and Management

This study was conducted from April 2018 to January 2019 on a commercial medium-sized dairy farm with approximately 90 lactating dairy cows in Ratchaburi province, Thailand. From a previous report of the Thai Meteorological Department, the experimental cows were classified into 2 groups according to their expected calving months, namely hot-rainy and cool periods. All healthy pregnant dry crossbred Holstein Friesian cows (Holstein Friesian with Thai native or Sahiwal breed) had an expected calving date in either one of the two periods. The hot and rainy group consisted of cows that calved from April 2018 to August 2018 (*n =* 29), including 11 primiparous and 18 multiparous cows (2nd–8th lactation); the average number (mean ± SD) of lactations in this group was 2.82 ± 2.18. The cool group consisted of cows that calved from November 2018 to January 2019 (*n =* 37). This group consisted of 13 primiparous and 24 multiparous cows, averaging 2.94 ± 2.36 lactations (Table 1).

The cows were raised in open barns with a loose housing system. They were fed commercial concentrates twice daily and had free access to roughage (grasses, corn stalks, corn leaves, and rice straw) and water. The cows were milked twice daily (at 5.30 a.m. and 4.30 p.m.) using the pipeline system. Electric fans under the roof were used during the afternoon until milking in the evening in the hot season.

### 2.3. Reproductive Management and Blood Sampling

The voluntary waiting period was established as 4 weeks after calving. Cows were examined by transrectal palpation and ultrasonography (Honda HS 1600 with a 7.5 MHz rectal probe, Honda, Tokyo, Japan) to measure ovarian structure and uterine involution. Cows with a corpus luteum (CL) and uterine involution structure were included. Cyclic cows were recruited into the experiment by purposive sampling with the consideration of farmers to avoid interfering with herd management. Selected cows received the PGF2α protocol: they were injected with 2 mL of PGF2α (Cloprostenol, Estrumate^®^, Intervet International B.V., Bangkok, Thailand) to regress the CL on day 0. Thereafter, the farmers observed estrous signs of the injected cow twice a day (morning and evening) by looking for standing to be mounted, increasing contact with other cows, or raising her tail head. The blood sample was taken in 2 periods. The first period of blood collection began one day after the injection of PGF2α. If a CL was not found on either ovary in the fourth week after calving, the veterinarian rechecked that cow using transrectal palpation and ultrasonography in the sixth and eighth weeks. Cows with CL at the sixth and eighth weeks also received an injection of PGF2α while cows without a CL were considered anestrous and excluded from the study. Subsequently, the second period of blood collection was performed 22 days after the injection of PGF2α, which was expected to be their spontaneous estrous cycle. The farmers observed estrous signs during this period twice daily. Cows that showed estrous signs after this spontaneous estrous cycle would be artificially inseminated approximately 12 h after the first estrous signs were detected and the insemination date was recorded. The cows were considered for artificial insemination until the end of the experimental period in each group (Table 1). All inseminated cows were observed until pregnancy confirmation at 30 days post-insemination by ultrasonography.

To monitor progesterone concentration, blood samples (5 mL) were taken daily from the jugular vein on day 1 after PGF2α injection and continuously until day 8 (i.e., blood samples were collected eight times), which was expected to include periods of early heat, standing heat, and after standing heat. In addition, blood samples were collected again eight times during the subsequent estrous cycle (days 22–29). After collection, all blood samples were left at room temperature for 30 min and then centrifuged (Eppendorf Centrifuge 5804 R, Humburg, Germany) with 1500× *g* for 10 min. The serum was harvested and stored at −20 °C until progesterone was assayed.

### 2.4. Rectal Temperature, Respiratory Rate, and Reproductive Performance Record

The rectal temperature (RT, in °C) was measured with a thermometer. RR (breaths per minute; bpm) was measured using a stethoscope with the ability to perform a physical examination. Both measurements were made prior to blood collection. The reproductive performance was explored and recorded by a veterinarian. The pregnant cow was confirmed by a veterinarian 30 days after artificial insemination by ultrasound. Cows with clinical illnesses were recorded and treated by a veterinarian.

### 2.5. Evaluation of Ultrasonography and Progesterone Concentration for Detection of Ovulation

The cows’ owners visualized estrous behaviors during the blood sampling period, and the estrous cows were conclusively confirmed by a veterinarian using transrectal palpation and ultrasonography on day 8 (the end of the first blood collection period) and day 29 (the end of the second blood collection period) following PGF2α injection. The presence of a suspected new CL was confirmed by the progesterone concentration (P4). If the hormone decreased to less than 1 ng/mL during the blood sampling period, it was defined as the day of regression of a mature CL in a cow [17,26] and was recorded as a new CL. Furthermore, the cows were also inseminated or showed estrous signs.

Progesterone concentration was measured by a secondary antibody enzyme immunoassay using the procedure of Brown et al. [27] in cooperation with the Endocrine Laboratory at the Khaokheow Open Zoo, Chonburi, Thailand. The intra-assay and inter-assay coefficients of variation in progesterone concentration were 2.53% and 4.95%, respectively.

### 2.6. Statistical Analysis

Statistical analysis was performed using the R program. Differences in mean THI, RR, RT, and calving to PGF2α injection, PGF2α injection to first service, and calving to first service (CTFS) intervals in each group were compared using Mann–Whitney tests or *t*-tests. The proportional data of estrus behavior with CL detection, the incidence of retained fetal membrane, and FSCR were analyzed using Chi-square tests. The distribution of the data was checked for normality using the Shapiro–Wilk test. Data were presented as means and standard deviation (SD). Analyses were considered statistically significant when *p* < 0.05.

## 3. Results

### 3.1. Temperature–Humidity Index and General Data

The mean THI for the hot and rainy season was 80.67 ± 0.79 (ranging from 79.80 to 81.89) and was defined as moderate HS. The mean monthly THI for the cool season was 77.81 ± 1.09 (range: 76.64–78.80) and was defined as mild HS (Table 1).

The number of cows selected under both moderate and mild HS conditions is presented in Table 2. At the beginning of the experiment, there were 29 cows in the moderate HS group, of which 22 returned to the estrous cycle. However, 13 of 22 cows returned within 4 weeks after calving and were included in the experiment. For the mild HS group, there were 37 cows at the start of the experiment, and 23 of 28 returned to the estrous cycle within 4 weeks after delivery. The occurrence of retained fetal membranes (RFMs) in the moderate HS group was 3.03 times higher than in the mild HS group. Furthermore, data on the incidence of cystic ovary (within 4 weeks postpartum) and the cause of culling (where necessary) are also shown in Table 2.

### 3.2. Effect of Heat Stress on Rectal Temperature and Respiratory Rate

The average RT and RR in moderate and mild HS are shown in Table 3. The average RR was significantly lower in the mild HS group than in the moderate HS group (*p* < 0.05). Furthermore, the average RT was also significantly lower for the mild HS group than for the moderate HS group (*p* < 0.05).

### 3.3. Effect of Heat Stress on Estrous Behavior and the Response of PGF2α in Cycling Cows

Progesterone concentrations of cows in both moderate and mild HS groups are presented in Figure 1. It was observed that the concentrations declined after PGF2*α* injection; thereafter, the concentrations seemed to increase 6 and 5 days post-injection in moderate and mild HS cows, respectively (Figure 1A). Interestingly, the decline in progesterone concentration patterns was observed approximately 26 days after PGF2*α* injection.

However, the patterns of progesterone concentrations during the 8 days of blood collection in the subsequent estrous cycle period seemed similar for cows in both moderate and mild HS groups (Figure 1B).

The 22 cows in each group received an injection of 2 mL PGF2α as shown in Table 2. It was noted that six cyclic cows in mild HS were not included in PGF2α treatment due to the farmer’s decision. Estrous behaviors and CL detection as responses to PGF2α confirmation and progesterone concentration (less than 1 ng/mL) were then investigated to classify the cow group. The group of PGF2α-induced cows was classified in that cycle and the subsequent estrous cycle, as shown in Table 4. After injection of PGF2α, the percentage of cows in estrus + new CL in mild HS was significantly higher than in moderate HS. Interestingly, the percentage of silent heat cows (without estrus + new CL) was higher under moderate than mild HS. Therefore, these findings indicated that the percentage of cows with a good response to PGF2α and estrus behavior (estrus + new CL) was mainly in mild HS conditions. The incidence of new CL was also detected in moderate HS cows without estrous behavior. As a consequence, the percentage of artificially inseminated cows (AI) did not differ between the two groups in the subsequent estrous cycle and the next estrous cycle. Two cows in the moderate HS group were culled due to the farmer’s decision. In the mild HS group, two cows were excluded from the study because they were inseminated out of the study plan and one cow was not inseminated during the observation period. Therefore, the 20 and 19 inseminated cows in moderate and mild HS were used for further observations.

The results of days from calving to PGF2α injection, PGF2α injection to first service, and calving to first service (CTFS), as well as the first-service conception rate (FSCR), are presented in Table 5. The average days from calving to PGF2α injection and CTFS of cows in mild HS were significantly shorter than cows in moderate HS (32.00 ± 2.11, 69.47 ± 18.18 vs. 42.50 ± 15.69, 84.60 ± 27.68), while the average days from PGF2α injection to first service of cows was not different between moderate and mild HS cows. Moreover, the FSCR of cows in mild HS tended to be higher than cows in moderate HS conditions (42.11% vs. 15%) (*p* = 0.06). Additionally, both primiparous and multiparous cows were used in this experiment (Table 1) and some of them were recorded as inseminated cows. Therefore, these findings indicated that HS suppresses reproductive performance in cows, especially in moderate HS conditions.

To clarify whether the incidence of RFM is caused by environmental factors such as HS, the association between HS and the incidence of RFM was measured in Table 6. The result showed an association between the HS group and the incidence of RFM, such that the incidence of RFM was higher in moderate HS than under mild HS conditions. Furthermore, 20 artificially inseminated cows in moderate HS were analyzed along with the occurrence of RFM (Table 7). This result indicated that there was no difference between CTFS from RFM and non-RFM cows in the moderate HS group. Therefore, RFM did not affect reproductive performance in this study.

## 4. Discussion

The value of THI was relatively high throughout the experimental period. The temperature gradually increased in April to reach the maximum level in May. It then slightly decreased until September and October, fell to its lowest at the end of 2018, and remained low until January 2019. In contrast, the average humidity was stable during the summer period (May–August), which saw high temperatures and low humidity. The humidity gradually increased in September to reach its highest level in October, which was the rainy season. Humidity then decreased from November to January (the winter season). The THI was similar to previous reports in Thailand [24,28]. Therefore, chronic HS occurred in cows raised in Thailand throughout the year, and the cows in this study were affected by moderate and mild HS during the experiment.

The RR and RT results were higher in moderate than in mild HS. According to the field study of Shehab-El-Deen et al. [10], cows had significantly higher RR and RT (95.5 ± 1.1 and 39.88 ± 0.06, respectively) in summer (when the THI was 82.10) than in winter (when THI was 66.40) (RR: 43.89 ± 0.61; RT: 38.94 ± 0.07). Furthermore, when cows were exposed to acute HS in heat chambers, it was found that RR increased immediately after heat exposure (76.02 ± 1.70 bpm) compared to the control (39.70 ± 0.71 bpm) [29]. Moreover, in this study, daily THI was moderately correlated with both RT (*r* = 0.42) and RR (*r* = 0.41), consistent with previous studies [30,31,32]. The increase in body temperature represented by RT would increase as a physiological mechanism in homeothermic animals, signifying a response in the cows from the induced heat load in their environment. The cows still showed a significant difference in RR and RT between moderate and mild levels of HS even though they were chronically exposed to HS. In this study, cows in moderate HS showed more adapted behavior to heat dissipation for an immediate return to the thermoneutral zone compared to those in mild HS. These results confirm that the cows in the two groups were affected by different levels of HS and had different general responses.

Thai dairy farmers usually set the voluntary waiting period to at least 60 days postpartum and begin to observe their cows for estrous behaviors. However, postpartum uterine involution is considered the key factor in setting the waiting period and necessary for the normal reproduction of dairy cows [33,34]. The recovery time of uterine involution was previously reported to be around 25 days postpartum in both multiparous and primiparous dairy cows [34]. Therefore, all healthy cows at 4 weeks of the voluntary waiting period in our study with confirmed complete involution of the uterine horn could be assumed to be normal postpartum cows that could initiate an estrous cycle. A short waiting period in our study (4 weeks) was used in order for the number of selected cows in this study to be sufficient to follow in each period, i.e., moderate HS and mild HS periods. In addition, this study design, with its short waiting period, would not interfere with the routine reproductive management set by the farmer.

The percentage of cows with a good response to PGF2α (estrus + new CL) was significantly higher in mild than in moderate HS conditions (*p* < 0.05). This difference could suggest that cows under mild HS were more responsive to PGF2α and showed estrous signs better than cows under moderate HS. However, the percentage of cows that had a new ovulation (estrus + new CL and no estrus + new CL groups) did not differ between moderate and mild HS (*p* = 0.75). Typically, the CL is regressed by PGF2α, after which the dominant follicle will ovulate. HS alters follicular development and steroid hormone synthesis [6]. The heat conditions affected the poor quality of the dominant follicle with low granulosa cells [19,35]. In this study, the development of the follicle during moderate HS might be qualitatively less than that during mild HS. Moreover, failure of heat detection could have resulted from either human error or silent heat, which is a typical problem in cows with HS. One of the possible causes for the reduction in estrous behavior in summer could be a reduction in estradiol level [22]. Low estradiol levels would not be sufficient to induce the LH (luteinizing hormone) surge, resulting in a prolonged estrous cycle in the summer period. Furthermore, the percentage of non-estrus + new CL cows in moderate HS (9/22, 40.91%) was significantly higher than in mild HS (3/22, 13.64%). Therefore, these findings were assumed to show that cows with mild HS responded to PGF2α administration and showed better follicular development and estrous signs than cows with moderate HS. Using hormone injections such as PGF2α (one or double injection protocol) could increase the estrous detection rate during HS conditions.

The number of artificially inseminated cows observed in the following estrous cycle did not differ between the two groups. Possible reasons for this result were that the number of new ovulations in the previous cycle did not differ because the hormone injections were sufficient to trigger ovulation equally well, even under moderate HS, or that temperature variation throughout the day could affect follicular development. The result of this study is similar to that of Pongpiachan et al. [36], who reported that the incidence of spontaneous estrous behavior after calving did not differ significantly in each season.

In this study, the reproductive index was concerned with FSCR and the calving to PGF2α injection, PGF2α injection to first service, and calving to first service (CTFS) intervals. All indices could be affected by the level of HS as a result of its effects on follicular development and the estrous cycle. These measures did not consider repeat-breeding cows and prolonged anestrus with many unknown factors. However, neither estrous indicators, the presence of the CL, nor the incidence of artificial insemination differed in this study. The FSCR under mild HS tended to be higher than in moderate HS (*p* = 0.06), and the average calving to PGF2α injection and CTFS intervals of cows under mild HS were significantly shorter than cows under moderate HS. During HS, altered hormonal levels have affected follicular development, delayed ovulation [6,19], and resulted in the inappropriate timing of fertilization between the oocyte and the sperm. Furthermore, HS reduces the potential for oocyte development [37]. Therefore, HS led to a lower FSCR [38,39] and a prolonged average CTFS [23]. Therefore, it might also directly affect the extension of open days (calving-to-conception interval) [24].

In this study, RFM was recorded in cows during moderate and mild HS (Table 2 and Table 6). The incidence of RFM was higher in moderate than in mild HS conditions, which aligns with previous studies [40,41,42]. HS increased reactive oxygen species and the immune response [7,43], causing atony of the uterus during calving [44]. This mechanism might increase the percentage of RFM during the summer and explain the results reported here.

## 5. Conclusions

HS influenced reproductive performance and general physiological responses in cross-bred Holstein Friesian dairy cows under tropical conditions in Ratchaburi province, Thailand. The average RT and RR of cows were more affected in moderate than mild HS. The FSCR of cows in mild HS tended to be higher than that of cows in moderate HS. The average CTFS of cows under mild HS was shorter than that of cows under moderate HS. Cows with mild HS had a relatively better response to PGF2α when compared to cows with moderate HS. This research might indicate that moderate HS rather than mild HS conditions have a significant impact on the reproductive performance of cows raised in tropical climates like Thailand across the year.

## Figures and Tables

**Figure 1 animals-14-02009-f001:**
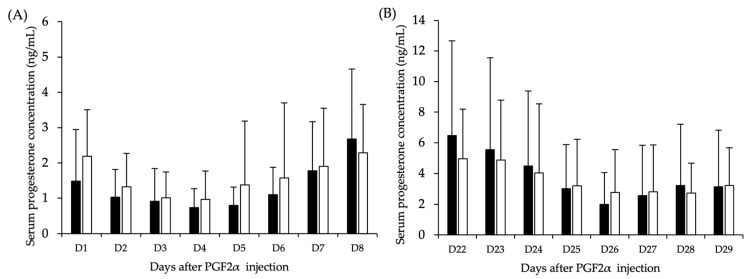
Serum progesterone concentrations of cows in moderate HS (closed bar) and mild HS (open bar) in (**A**) the first (D1–8) and (**B**) second (D22–29) periods of blood collection after PGF2α injection. Data are represented as means and the error bars are SDs.

**Table 1 animals-14-02009-t001:** Description of the period of the experiment and temperature–humidity index (THI) and data of cows’ body condition score (BCS), lactation number, and milk yield. Data represented as mean ± SD.

Parameter	Moderate HS Group (*n =* 29)	Mild HS Group (*n =* 37)
Period of experiment	April–August 2018	November 2018–January 2019
THI	80.67 ± 0.79	77.81 ± 1.09
BCS ^1^	3.34 ± 0.38	3.22 ± 0.45
Lactation number	2.82 ± 2.18	2.94 ± 2.36
Milk yield ^2^	18.32 ± 2.13	19.66 ± 2.12

^1^ Body condition scores (BCSs) of cows were measured 4 weeks before calving. ^2^ Milk yields of cows were recorded 4 weeks after calving.

**Table 2 animals-14-02009-t002:** Distribution of the cows with general data.

Parameter	Moderate HS Group	Mild HS Group
Total sample size	29	37
Cyclic cows	22	28
Cyclic cows within 4 weeks	13	23
Cyclic cows during 4 to 8 weeks	9	5
Retained fetal membranes	12	7
Cystic ovary ^1^	4	2
Culling	3	1
Anaplasmosis	1	0
Cystic ovary > 30 days	1	1
Hip luxation	1	0
Cyclic cows received an injection of PGF2α ^2^	22	22

^1^ Cystic ovary within 4 weeks postpartum (spontaneous recovery). ^2^ PGF2α = Estrumate^®^ (cloprostenol sodium).

**Table 3 animals-14-02009-t003:** Maximum, minimum, and mean respiratory rates (RRs) and rectal temperature (RT) in moderate (*n =* 22) and mild HS (*n =* 22) groups. Data represented as mean ± SD.

Parameter	HS Group	Maximum	Minimum	Mean
RT (°C)	Mild	38.96 ± 0.22 ^a^	37.78 ± 0.78 ^a^	38.47 ± 0.21 ^a^
	Moderate	39.45 ± 0.52 ^b^	38.38 ± 0.20 ^b^	38.87 ± 0.15 ^b^
RR (bpm)	Mild	48.55 ± 7.54 ^a^	33.45 ± 3.16 ^a^	41.04 ± 4.55 ^a^
	Moderate	62.82 ± 13.48 ^b^	39.82 ± 9.76 ^b^	51.17 ± 10.52 ^b^

^a–b^ Different letters in the same column indicate significant differences (*p* < 0.05) between groups. bpm = breaths per minute.

**Table 4 animals-14-02009-t004:** Distribution of the cows with estrous behavior and new corpus luteum (CL) in the PGF2α-induced cycle and the number of first artificially inseminated cows after the subsequent estrous cycle.

Parameter	Moderate HS Group (*n =* 22)	Mild HS Group (*n =* 22)
Estrous cycles induced by PGF2α		
Estrus + new CL ^1^	18.18% (4/22) ^a^	54.55% (12/22) ^b^
Estrus + No new CL ^2^	13.64% (3/22)	13.64% (3/22)
No Estrus + new CL ^3^	40.91% (9/22) ^a^	13.64% (3/22) ^b^
No Estrus + No new CL ^4^ (non-response)	27.27% (6/22)	18.18% (4/22)
After subsequent estrous cycle		
First AI in subsequent estrous cycle ^5^	40.91% (9/22)	50.00% (11/22)
First AI in 2nd subsequent estrous cycle ^6^	27.27% (6/22)	13.64% (3/22)
First AI in 3rd or 4th subsequent estrous cycle ^7^	22.72% (5/22)	22.72% (5/22)
No AI ^8^	-	4.55% (1/22)
Other events ^9^	-	9.09% (2/22)

^1^ PGF2α-induced cows showed estrous behavior, and a new CL was found. ^2^ PGF2α-induced cows showed estrous behavior, and a new CL was not found. ^3^ PGF2α-induced cows did not show estrous behavior, but the level of progesterone was ≤1 ng/mL and/or a new CL was found. ^4^ Cows did not respond to PGF2α and were excluded from the study. ^5^ Cows were first artificially inseminated in the first subsequent estrous cycle. ^6^ Cows were first artificially inseminated in the second subsequent estrous cycle. ^7^ Cows were first artificially inseminated in the third or fourth subsequent estrous cycle. ^8^ A cow was not inseminated during the observation period. ^9^ Cows were first artificially inseminated after PGF2α injection and were excluded from the study. ^a–b^ Different letters in the same row indicate significant differences (*p* < 0.05) between groups.

**Table 5 animals-14-02009-t005:** Comparison of average days from calving to PGF2α injection, PGF2α injection to first service, calving to first service (CTFS), and first-service conception rate (FSCR) between moderate and mild HS groups. Data represented as mean ± SD.

Parameter	Moderate HS Group (*n =* 20)	Mild HS Group (*n =* 19)
Days from calving to PGF2α injection	42.50 ± 15.69 ^a^	32.00 ± 2.11 ^b^
Days from PGF2α injection to first service	42.10 ± 19.03 ^a^	37.47 ± 17.55 ^a^
Days from calving to first service (CTFS)	84.60 ± 27.68 ^a^	69.47 ± 18.18 ^b^
FSCR (%)	15 (3/20) ^a^	42.11 (8/19) ^a^

^a–b^ Different letters in the same row indicate significant differences (*p* < 0.05) between groups.

**Table 6 animals-14-02009-t006:** Association between moderate (*n =* 22) or mild HS (*n =* 22) and incidence of retaining the fetal membrane (RFM) in cows. Data represented as mean ± SD.

Parameter	Moderate HS Group	Mild HS Group	Chi-Square Value	*p*
RFM cow ^1^	9	3	4.125	0.042 *
Non-RFM cow ^2^	13	19		

* Significant association (*p* < 0.05). ^1^ Cows failed to expel fetal membranes within 24 h after calving. ^2^ Cows did not fail to expel fetal membranes within 24 h after calving.

**Table 7 animals-14-02009-t007:** Comparison of the average days from calving to first service (CTFS) between cows that retained the fetal membrane (RFM) and those that did not in moderate HS (*n =* 20). Data represented as mean ± SD.

Parameter	RFM Cow (*n =* 8)	Non-RFM Cow (*n =* 12)
Days from calving to first service (CTFS)	94.25 ± 36.60 ^a^	78.17 ± 18.92 ^a^

^a^ Similar letters in the same row indicate insignificant difference (*p* ≥ 0.05) between groups.

## Data Availability

The data that support the findings of this study are available upon request from the corresponding author.

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
