# Peer review of "Effect of Heat Stress on Subsequent Estrous Cycles Induced by PGF2α in Cross-Bred Holstein Dairy Cows"

_animals, 2024, doi:10.3390/ani14132009_

Round 1

Reviewer 1 Report

Comments and Suggestions for Authors

This study attempts to explain the Effect of heat stress on subsequent estrous cycles induced by 2 PGF2α in cross-bred Holstein dairy cows raised in Ratchaburi 3 province, Thailand”. The experiment presents several caveats; the most important are the age of the cows (range 2 to 8) and the waiting period for cycling/insemination. These variables cannot fix, and therefore, the manuscript is rejected.

I listed my other concerns below in the order I found them in the manuscript.

Simple Summary

-        L15 – L16. HS? It is not defined.

Abstract

The abstract is missing important information regarding the treatments and cows.

-        L20. Are these crossbred animals? If so, please indicate it.

-        L21 – L22. Were cows similar in age/parity/body condition score? Were these cows selected after parturition? Please indicate it. Briefly describe the moderate and mild temperatures. How can authors ensure these temperatures? Indoor facilities?

-        L22 – L23. How were cycling cows detected? Were all at the same stage of the estrous cycle?

-        L27 – L28. Better response. Based on what? The authors can delete this sentence and save space for other information.

-        L31 – L32. Please indicate the statistical difference.

-        L32 – L34. I missed the novelty of the study.

Introduction

-        L49 – L50. What type? Severe? Moderate?

-        L68 – L74. I do not see the innovation. Why would the study differ in Thailand from those previously reported in other countries?

M&M

The authors need to provide a table with the THI, max and min temperatures per month.

-        L96 – L99. Please edit the sentence. It is not clear what the authors tried to say.

-        L99 – L104. This is a big caveat. Why did the authors not use cows from the same parity? Cows with different parities have different metabolisms and immune systems, which leads to different responses to chemical treatments. How did the authors correct this variation?

-        L105 – L109. Please indicate if this was for both treatments. Were the fans used in both seasons?

-        L112. Based on what? The recommendation is 60 d. It is recommended that it be extended in the summer (https://doi.org/10.3168/jds.2021-20707). Experience (age/parity) influences the extension period (https://doi.org/10.3168/jds.S0022-0302(90)78798-X; PMID: 31308594; https://doi.org/10.3168/jds.2022-22773). There is not much evidence supporting shortening the waiting period below 60 days. Uterine involution takes about 42 days in primiparous cows and 50 days in multiparous cows to occur. Breeding cows before this happens is bound to be unsuccessful.

o   https://doi.org/10.3168/jds.S0022-0302(07)71645-4

o   https://doi.org/10.3168/jds.2021-20707

o   https://doi.org/10.3168/jds.2010-3790

o   https://doi.org/10.3168/jds.S0022-0302(90)78798-X

o   PMID: 31308594

o   https://doi.org/10.3168/jds.2022-22773

-        L142 – L143. Were cows inseminated only one time? Or do they have more than one opportunity? How many attempts per animal? Per age?

-        L143 – L144. How many cows were presented with clinical illnesses? Were they kept in the study?

-        L160 – L166. How did the authors correct for age? How did the authors consider the fan effect?

-        L163 – L164. The Retained placenta membrane was not part of the experiment's objectives. Why is this variable included in the analyses?

Results

The authors cannot expect the same response from a 2-parity cow vs. a 4/5 parity cow or 8 8-parity cow. They need to separate the effects. The cows were crossbred and possibly adapted to the conditions of the location. Why did the authors not measure any other hormone with a better understanding of heat stress (cortisol or catecholamines)? It is possible that the response was not due to the heat stress, but the waiting period. Therefore, the experimental design is incorrect.

Reviewer 2 Report

Comments and Suggestions for Authors

this study dealing with a serious problem that affect alot of farmers around the world. so, this is very inters topic,

how ever, the current study cosist of very low number of cows and those the result an statistic are doutable.

i am missing any data on colling and ventilation . did you have colling facilities or not? 

rectal palpation need to be measure during all day today you can use automatic device inserted to cow vagine for 5 days, giving you temprature reding every 10 min.

result: there is no explanation to the finding. insted you put it in the disscusion. you need to talk about the result and presenet the data in the result section. in the disscusion you need just to disscus it without the data and compare to other manuscripts.

you not suppose to give all the numbers and Pvalue in the disscusion for example see line: 267-269.

Reviewer 3 Report

Comments and Suggestions for Authors

Thank you for your effort in conducting the study on the effect of heat stress on subsequent estrous cycles induced by PGF2α in cross-bred Holstein dairy cows raised in Ratchaburi province, Thailand. The findings hold potential significance for the scientific community. However, I have a few queries regarding specific areas that need addressing to maximize the manuscript's suitability for publication.

1. The addition of “raised in Ratchaburi province, Thailand” in the manuscript title may be unnecessary.

2. The background of the study appears to be somewhat lacking in depth. I recommend strengthening the background section to enhance the context and relevance of the research.

3. The experiment’s data is nearly five years old. Considering the annual fluctuations in global temperatures, how does the author justify the relevance of this experiment to the current cattle industry?

Addressing these points will likely enhance the manuscript's quality and increase its chances of acceptance.

Round 2

Reviewer 2 Report

Comments and Suggestions for Authors

Thank you for answering my questaines.

Author Response

Dear the Reviewer,

Thank you very much for accepting our responses on your kind comments. We appreciate your kind review on our manuscript.

Best regards,

Theera Rukkwamsuk